# The Association between Iron Deficiency and Renal Outcomes Is Modified by Sex and Anemia in Patients with Chronic Kidney Disease Stage 1–4

**DOI:** 10.3390/jpm13030521

**Published:** 2023-03-14

**Authors:** Pei-Hua Yu, Yu-Lin Chao, I-Ching Kuo, Sheng-Wen Niu, Yi-Wen Chiu, Jer-Ming Chang, Chi-Chih Hung

**Affiliations:** 1Division of Nephrology, Department of Internal Medicine, Kaohsiung Medical University Hospital, Kaohsiung Medical University, Kaohsiung 80708, Taiwan; 2Division of Nephrology, Department of Internal Medicine, Kaohsiung Municipal Ta-Tung Hospital, Kaohsiung Medical University, Kaohsiung 80145, Taiwan; 3Regenerative Medicine and Cell Therapy Research Center, Kaohsiung Medical University, Kaohsiung 80708, Taiwan

**Keywords:** chronic kidney disease, iron, transferrin saturation, sex, anemia, renal outcomes

## Abstract

Iron deficiency is prevalent in women and patients with chronic kidney disease (CKD). Iron deficiency is not only related to anemia but contributes to adverse consequences for the kidney as well. Whether iron status is associated with renal outcomes after considering sex and anemia in patients with CKD stage 1–4 is unclear. Thus, we investigated the association of iron or iron saturation with renal outcomes in a CKD cohort. During a follow-up of 8.2 years, 781 (31.2%) patients met the composite renal outcome of renal replacement therapy and a 50% decline in renal function. In linear regression, iron was associated with sex, hemoglobin (Hb), and nutritional markers. In a fully adjusted Cox regression model, the male patients with normal iron had a significantly decreased risk of renal outcomes (hazard ratio (HR) 0.718; 95% confidence interval (CI) 0.579 to 0.889), but the female patients did not exhibit this association. The non-anemic patients (Hb ≥ 11 g/dL) had a decreased risk of renal outcomes (HR 0.715; 95% CI 0.568 to 0.898), but the anemic patients did not. In the sensitivity analysis, transferrin saturation (TSAT) showed similar results. When comparing iron and TSAT, both indicators showed similar prognostic values. In conclusion, iron deficiency, indicated by either iron or iron saturation, was associated with poor renal outcomes in the male or non-anemic patients with CKD stage 1–4.

## 1. Introduction

Iron is a unique trace nutrient that participates in numerous fundamental biological reactions [1,2,3]. It is not only crucial for the synthesis of heme, which is responsible for oxygen delivery to the tissues, but also serves as an indispensable element of the iron-sulfur cluster that constitutes a wide range of proteins or enzymes involving mitochondrial energetics and DNA synthesis and repair, as well as cellular differentiation and growth [1,4,5]. As a vital component of hematopoiesis, iron deprivation pertains to anemia, which gives rise to cardiovascular morbidity, poor health-associated quality of life, and increased all-cause mortality [6,7,8]. Moreover, owing to its essential role in energy metabolism, the insufficiency of iron contributes to adverse consequences for organs with high energy demands, such as the brain, heart, immune-competent cells, and skeletal muscle [3,9]. The clinical manifestations of iron deficiency are cognitive dysfunction [2,10,11,12], exacerbation of heart failure [9,13,14], impaired immune function [15], and decreased physical performance [1,15]. However, there is limited evidence exploring the effect of iron on renal function, even though the kidney is thought to possess high energy requirements [16,17]. Del Greco et al. indicated that iron has a beneficial effect on kidney function in the general population [18]. Nevertheless, the relationship between iron status and renal function in the group of non-dialysis-dependent chronic kidney disease (CKD) patients remains uncertain.

Sex and anemia are two important confounders to be considered. Prior studies suggest that women have a higher prevalence of CKD while men progress at a faster rate of renal function decline [19,20]. The possible mechanisms of sex dimorphism in CKD could be attributed to the renal structure, differing hemodynamic response, and influence of sex hormones [21,22,23]. Iron deficiency is common in females and is caused by reduced dietary iron intake and blood loss during menstruation [2,24]. Nonetheless, sex differences in the association between iron status and renal outcomes in CKD patients have not been fully studied.

Anemia is prevalent in CKD patients. Patients with hemoglobin (Hb) <11 g/dL made up 32.2% of our cohort. Deficiency of erythropoietin and deficiency of iron are the two major reasons for anemia in CKD [25]. Iron-deficiency anemia (IDA) could be due to a true paucity of iron stores (absolute IDA) and relative (functional) iron deficiency. A better iron status could overcome the risk of functional iron deficiency [26]. Furthermore, an erythropoiesis-stimulating agent (ESA) was given late to our cohort (when the serum creatinine was >6 mg/dL). Anemia in these patients could be a risk for renal function progression [6,27]. Whether iron deficiency is still important in patients with anemia is still unclear.

Accordingly, the purposes of our study were to explore the relationship between iron status and renal outcomes, as well as to investigate the impact of sex and anemia in this association among patients with CKD stage 1–4. Considering that the conventional iron biomarker transferrin saturation (TSAT) is susceptible to malnutrition and inflammation [28], we utilized serum iron as the marker of iron status in our main analysis and TSAT in the sensitivity test.

## 2. Materials and Methods

### 2.1. Study Design and Participants

We conducted a prospective, observational cohort study in two affiliated hospitals (Kaohsiung Medical University Hospital and Kaohsiung Municipal Hsiao-Kang Hospital) of Kaohsiung Medical University in Southern Taiwan. There were 3659 CKD stage 1–5 patients enrolled in the Integrated CKD Care Program Kaohsiung for Delaying Dialysis from 11 November 2002 to 31 May 2009. CKD was staged according to the definition of the K/DOQI guidelines [29], and the estimated glomerular filtration rate (eGFR) was calculated using the equation from the 4-variable Modification of Diet in Renal Disease (MDRD) study. Because the use of ESA could increase the Hb level even in those with low iron status, the patients with CKD stage 5 were excluded. Hence, the final study cohort contained 2500 CKD stage 1–4 patients. All the participants in our study provided informed consent, and the protocol followed in the study was approved by the Institutional Review Board of Kaohsiung Medical University Hospital, with the study being performed according to the approved guidelines. The follow-up period was 3000 days (approximately 8.2 years).

To study the impact of serum iron and sex on renal outcomes, we stratified these 2500 CKD subjects into 4 groups: “men with normal iron”, “men with low iron”, “women with normal iron”, and “women with low iron”, in accordance with the tertiles of the serum iron levels; that is to say, the patients in the first tertile of serum iron (men < 67 μg/dL; women < 54 μg/dL) were assigned to the low iron groups, while those in the second and the third tertiles of serum iron (men ≥ 67 μg/dL, women ≥ 54 μg/dL) were allocated to the normal iron groups. In the sensitivity analysis, to study the impact of TSAT and sex on renal outcomes, we divided the participants into another four groups: “men with normal TSAT”, “men with low TSAT”, “women with normal TSAT”, and “women with low TSAT”, according to the tertiles of TSAT; that is, subjects in the first tertile of TSAT (men < 24.6%; women < 19.7%) were taken as low TSAT, while those in the second and the third tertiles of TSAT (men ≥ 24.6%; women ≥ 19.7%) were seen as normal TSAT.

In addition, to study the influence of serum iron and anemia on renal outcomes, the patients were grouped in line with the level of both serum iron and Hb. Individuals with Hb < 11 g/dL were classified as anemic (according to the definition in the study of the Trial to Reduce Cardiovascular Events with Aranesp Therapy [TREAT] [30]), while those with Hb ≥ 11 g/dL were regarded as non-anemic. Similarly, low or normal iron groups were defined by the first tertile of serum iron levels (67 μg/dL in men, 54 μg/dL in women) in the main analysis and by the first tertile of TSAT levels (24.6% in men, 19.7% in women) in the sensitivity test.

Furthermore, we studied the correlation of the combined set of serum iron and TSAT with renal outcomes. In the same way, tertile 1 of the serum iron levels was used to partition the subjects into low iron and normal iron groups, while tertile 1 of the TSAT levels was utilized to group the patients into low TSAT and normal TSAT groups.

### 2.2. Demographic, Medical, Lab Data, and Clinical Parameter Collection

Baseline parameters were acquired from medical records and visits with patients at enrollment. It included demographic features (age and sex); medical histories (diabetes mellitus (DM), cardiovascular disease (CVD), hypertension (HTN), hyperuricemia, metabolic syndrome (MtS), and malnutrition–inflammation); clinical parameters (mean arterial blood pressure (MAP), body mass index (BMI), and malnutrition–inflammation score (MIS)); and laboratory data (serum creatinine, Hb, albumin, white blood cell count (WBC), C-reactive protein (CRP), bicarbonate, phosphorus, calcium, uric acid, glycosylated hemoglobin (HbA1c), and urine protein to creatinine ratio (UPCR)).

The medical histories were collected by chart review by two to three nephrologists; DM and HTN were defined by clinical diagnosis. CVD was defined as a clinical diagnosis of ischemic heart disease, congestive heart failure, and cerebrovascular disease. Hyperuricemia was defined as uric acid > 7.2 mg/dL in men, >6.5 mg/dL in women, or that under urate-lowering therapy. MtS was defined by the meeting of at least 3 of the following 5 criteria proposed by the Health Promotion Administration, Ministry of Health and Welfare of Taiwan: (1) fasting blood glucose ≥ 100 mg/dL or previously diagnosed DM; (2) systolic blood pressure ≥130 mmHg or diastolic blood pressure ≥85 mmHg or a previous diagnosis of HTN; (3) high-density lipoprotein cholesterol < 40 mg/dL in men or <50 mg/dL in women; (4) triglyceride ≥ 150 mg/dL; and (5) waist circumference ≥ 90 cm in men or ≥80 cm in women [31].

The MIS, first proposed by Professor Kalantar-Zadeh for dialysis patients [32] and modified for CKD patients, was composed of ten items, including body weight change, dietary intake, gastrointestinal symptoms, nutritionally related functional impairment, comorbidity, fat stores, signs of muscle wasting, BMI, albumin, and total iron-binding capacity (TIBC). Each item was classified according to 4 levels of severity, from 0 (normal) to 3 (severely abnormal), with the total score ranging from 0 to 30. Malnutrition–inflammation was defined by an MIS of >4 points, based on the receiver operating characteristic curve for outcome prediction.

Ferritin was measured in serum using a two-site immuno-enzymatic assay with the Beckman Coulter UniCel-DxI 800 (Beckman Coulter Inc., CA, USA). Iron was measured in serum using an Fe reagent via a timed endpoint method with the Beckman Coulter UniCel-DxC 800 (Beckman Coulter Inc., CA, USA). Transferrin was measured in serum using a transferrin reagent via the turbidimetric method with the Beckman Coulter UniCel-DxC 800 (Beckman Coulter Inc., CA, USA). Additionally, TIBC values were obtained by multiplying the transferrin (mg/dL) by a coefficient of 1.4. High-sensitivity CRP was measured in serum using the near-infrared particle immunoassay rate method with the Beckman Coulter UniCel-DxC 800 (Beckman Coulter Inc., CA, USA). The laboratory data were averaged and analyzed three months before and after enrollment in the CKD care system.

### 2.3. Outcomes

There were 2 main outcomes assessed: (1) renal replacement therapy (RRT) and (2) the composite endpoint of RRT and a 50% decline in eGFR. RRT was defined as the entry into long-term dialysis therapy (hemodialysis or peritoneal dialysis) or the receiving of kidney transplantation. For the outcome of RRT, the subjects were censored due to death or loss of follow-up.

### 2.4. Statistical Analysis

The statistical results of the baseline characteristics of the patients were expressed as mean ± standard deviation (SD) for the continuous variables with normal distribution, as median (25th, 75th percentile) for the continuous variables with skewed distribution, and as percentages for the categorical data.

The significance of the differences for the normally distributed continuous variables among the groups was tested using one-way ANOVA analysis, while the difference for the non-normally distributed continuous variables was tested using a Kruskal–Wallis analysis. Additionally, the difference in the distribution of categorical variables among groups was tested using the chi-square test. Cox proportional hazard regression models were performed, in separate groups of males and females, to assess the association of serum iron with the outcomes, the relationship of TSAT with the outcomes, and the correlation of the combined set of serum iron and TSAT with the outcomes. The adjusted hierarchical covariates were as follows: model 1 adjusts for age, eGFR, and log-transformed UPCR; model 2 adjusts for the covariates in model 1 plus DM, CVD, severe liver disease, cancer, mean blood pressure (BP), and BMI; and model 3 adjusts for the covariates in model 2 plus albumin, log-transformed CRP, Hb, log-transformed ferritin, phosphorus, total cholesterol, and HbA1c. Statistical analyses were conducted using SPSS 21.0 for Windows (SPSS Inc., Chicago, IL, USA), with statistical significance set at a two-sided *p*-value of <0.05.

## 3. Results

### 3.1. Baseline Characteristics of Participants

In total, 2500 CKD stage 1–4 patients (1601 men and 899 women) were included and analyzed in this study. The baseline demographics, medical histories, clinical parameters, and laboratory indices are detailed in Table 1.

Among the male participants, the mean age (±SD) was 62.8 ±14.4 years; 764 (47.7%) had DM; 365 (22.8%) had a history of CVD; 957 (59.8%) had HTN; and 370 (23.1%) suffered from malnutrition–inflammation. The overall levels of eGFR and Hb were 38.2 mL/min/1.73 m^2^ (interquartile range (IQR): 26.7–49.7 mL/min/1.73 m^2^) and 12.8 ± 2.2 g/dL, respectively. During the follow-up of 8.2 years, there were 313 (19.6%) male patients starting RRT; 432 (27.0%) met the composite endpoint of RRT combined with a 50% decline in eGFR, and 274 (17.1%) expired before RRT. In contrast to the male subjects with normal iron, those with low iron tended to have DM, CVD (particularly congestive heart failure and cerebrovascular disease), malnutrition–inflammation, higher MIS, lower eGFR, higher UPCR, lower Hb, decreased albumin, elevated WBC, elevated CRP, lower bicarbonate, and higher HbA1c. Likewise, the male patients in the low iron group were likely to have poor renal outcomes and higher mortality before RRT compared to those in the normal iron group.

In the female participants, those with low iron tended toward DM and malnutrition–inflammation, followed by higher MIS, higher UPCR, lower Hb, lower albumin, elevated WBC, elevated CRP, and elevated HbA1c in comparison with those with normal iron. Moreover, the female patients in the low iron group were inclined to receive RRT and had higher mortality before RRT in contrast to those in the normal iron group.

In summary, both the male and the female CKD stage 1–4 patients with low iron levels were likely to suffer from DM and malnutrition–inflammation. In addition, worse baseline renal function (lower eGFR or higher UPCR), more severe anemia, declines in nutritional status (lower albumin), inflammation (including elevated WBC and CRP levels), poor renal outcomes, and higher mortality before RRT were disclosed in the low iron subjects of both sexes. Notably, higher values of WBC, CRP, and MIS were present in the low iron group of men in comparison to the low iron group of women. With regard to Hb levels, the women’s Hb levels were lower overall than the men’s, regardless of the iron status in both sexes.

### 3.2. Iron Biomarkers in the Separate Sex Groups and Their Relationships with Anemia, Malnutrition, and Inflammation

The iron status of the study participants is depicted in Table 2. Only 0.8% of the patients received iron therapy, and no patient in the study was treated with ESA as reimbursement by the Taiwan National Health Insurance is disallowed. Generally, the overall iron parameters, such as serum iron, TSAT, and ferritin, of the women were lower than those of the men, while the TIBC of the females was higher than that of the males.

Furthermore, Table 2 also shows that the patients with low iron status in both the male and the female groups were prone to hypoalbuminemia (albumin < 3.5 g/dL), elevated CRP values (CRP > 3 mg/L), anemia (Hb < 11 g/dL), and malnutrition–inflammation (MIS > 4 points). In particular, the subjects in the low iron group of men had the highest percentage of elevated CRP levels compared with the other three groups. Additionally, there was a higher percentage of anemia in both the normal iron and the low iron groups of women in comparison with that of the men.

### 3.3. Association of Iron Profiles with Adverse Renal Outcomes Divided by Sex

Table 3 shows the results of the Cox regression analysis for the adverse renal outcomes.

For the male patients, one SD increase in serum iron was associated with a 16.2% decrease in RRT (hazard ratio (HR) 0.838; 95% confidence interval (CI) 0.717 to 0.980), along with an 11.8% decrease in RRT and a 50% decline in eGFR (HR 0.882; 95% CI 0.778 to 0.998). On the other hand, for the female patients, there were no significant relationships between serum iron and adverse renal outcomes in the fully adjusted model.

As for the pre-specified groups of low iron and normal iron, in comparison to the male subjects with low iron, those with normal iron had a significantly reduced risk of RRT of 36.0% (HR 0.640; 95% CI 0.500 to 0.820), accompanied by a 28.2% reduction in the risk of the composite endpoint of RRT combined with a 50% decline in eGFR (HR 0.718; 95% CI 0.579 to 0.889) in the fully adjusted model. By contrast, the female patients with normal iron did not present a significantly decreased risk of adverse renal outcomes compared to those with low iron in the fully adjusted model.

Similarly, the negative association between TSAT and adverse renal outcomes was only exhibited in the males, regardless of whether TSAT was analyzed as a continuous variable or a category variable (Table 4). For the men, a 1 SD increase in TSAT was related to a 15.3% decrease in RRT (HR 0.847; 95% CI 0.734 to 0.978) as well as a 13.0% decrease in RRT and a 50% decline in eGFR (HR 0.870; 95% CI 0.774 to 0.978). Additionally, compared to the men with low TSAT, those with normal TSAT had a 33.8% decrease in the risk of RRT (HR 0.662; 95% CI 0.515 to 0.852) as well as a 23.0% decrease in the risk of RRT combined with a 50% decline in eGFR (HR 0.770; 95% CI 0.620 to 0.957). Nevertheless, the women did not show any significant relationship between TSAT and renal outcomes.

### 3.4. Association of Iron Profiles with Adverse Renal Outcomes Divided by Hb

As per the results demonstrated in Table 5, among the subjects without anemia (as Hb ≥ 11 g/dL), a 1 SD increase in serum iron demonstrated a 17.9% reduction in the risk of RRT (HR 0.821; 95% CI 0.687 to 0.980) as well as a 14.6% decrease in RRT combined with a 50% decline in eGFR (HR 0.854; 95% CI 0.749 to 0.975). Conversely, among those with anemia (Hb < 11 g/dL), there were no significant correlations between serum iron and adverse renal outcomes in the fully adjusted model.

Regarding the groups of low iron and normal iron, the non-anemic patients with normal iron had significantly decreased risks of RRT and the composite endpoint of RRT combined with a 50% decline in eGFR, with HRs (95% CI) of 0.699 (0.525 to 0.929) and 0.715 (0.568 to 0.898), respectively, compared to those with low iron in the fully adjusted model. Conversely, the anemic patients with normal iron did not present a significantly decreased risk of adverse renal outcomes in comparison to those with low iron in the fully adjusted model.

Likewise, the negative association between TSAT and adverse renal outcomes was only exhibited in the non-anemic subjects (Table 6). For the non-anemic patients, a 1 SD increase in TSAT correlated with a 15.9% reduction in the risk of RRT (HR 0.841; 95% CI 0.708 to 0.999) as well as a 14.2% decrease in RRT combined with a 50% decline in eGFR (HR 0.858; 95% CI 0.756 to 0.973). In comparing the non-anemic patients with normal TSAT to those with low TSAT, those with normal TSAT tended to have a reduced risk of RRT of 24.8% (HR 0.752; 95% CI 0.560 to 1.011) and a reduced risk of RRT combined with a 50% decline in eGFR of 20.4% (HR 0.796; 95% CI 0.630 to 1.006), despite the lack of statistical significance.

### 3.5. Association of “Combined TSAT and Serum Iron” with Adverse Renal Outcomes in Male CKD Stage 1–4 Subjects

We further separately divided the male subjects into four groups according to serum iron and TSAT levels, as Table 7 indicates. Compared with the reference group (i.e., “low iron low TSAT” group), the patients in the “normal iron normal TSAT” group had a significantly reduced risk of RRT of 39.7% and reduced risk of 28.7% for the composite endpoint of RRT plus a 50% decline in eGFR based on HRs (95% CI) of 0.603 (0.452 to 0.804) and 0.713 (0.556 to 0.914), respectively, in the fully adjusted model.

### 3.6. Factors Associated with Serum Iron

We further conducted multivariate linear regression to analyze the correlation factors of serum iron. As Appendix A reveals, the female subjects, DM patients, and those with a higher mean BP were significantly associated with lower serum iron, with β coefficients (95% CI; *p*-value) of −0.238 (−0.324 to −0.152; *p* < 0.001), −0.282 (−0.365 to −0.198; *p* < 0.001), and −0.005 (−0.008 to −0.003; *p* < 0.001), respectively. In contrast, the patients with higher Hb, albumin, and phosphorus were prone to having a higher serum iron, with β coefficients (95% CI; *p*-value) of 0.123 (0.101 to 0.146; *p* < 0.001), 0.150 (0.051 to 0.249; *p* = 0.003), and 0.075 (0.025 to 0.124; *p* = 0.003), respectively.

Moreover, this table also displays different degrees of negative correlation between the MIS and serum iron in both sexes. For the men, a higher MIS was significantly associated with lower serum iron (β coefficient −0.052; 95% CI −0.078 to −0.025; *p* < 0.001), but for the women, the MIS did not have a significant statistical correlation with serum iron (βeta coefficient −0.003; 95% CI −0.032 to 0.025; *p* = 0.807).

## 4. Discussion

In this observational study, the major finding was that sex and anemia modified the association between iron status and renal outcomes among patients with CKD stage 1–4. Moreover, serum iron and TSAT were negatively correlated with poor renal outcomes (RRT or RRT plus a 50% decline in eGFR) in the males and non-anemic patients with CKD stage 1–4. Therefore, both serum iron and TSAT were important predictors. In our cohort, the male patients in the first tertile of iron (<67 μg/dL) or in the first tertile of TSAT (<24.6%) were associated with a higher risk of poor renal outcomes, while the male patients with both iron ≥ 67 μg/dL and TSAT ≥ 24.6% had a lower risk for poor renal outcomes.

Iron deficiency, which affects up to 25% of the general population, is one of the most common nutritional disorders worldwide [24,33], occurring in up to 30–45% of CKD groups [34]. However, few studies have focused on the impact of iron status on renal function decline in CKD subjects until now. Previous studies have primarily delineated the association of abnormal iron balance with all-cause mortality, adverse cardiac events, and poor physical health-related quality of life [5,8,33,34,35,36]. To the best of our knowledge, this is the first study to demonstrate the correlation between iron status and renal outcomes in CKD populations.

There are two forms of iron deficiency. Apart from absolute iron deficiency, reflecting the paucity of iron stores in the bone marrow, a certain percentage of CKD patients suffer from functional iron deficiency representing reduced iron bioavailability despite adequate body iron storage [13,37,38]. The gold standard for the diagnosis of iron deficiency is a bone marrow biopsy, which represents absolute iron deficiency in the absence of iron in marrow fragments or macrophages, whereas functional iron deficiency is diagnosed in the absence of iron in erythroblasts [39]. Nevertheless, there are many limitations to bone marrow biopsy existing in clinical practice, including the invasive characteristics of the procedure, the high risk of complications, insufficient bone marrow samples, and the high subjectivity of marrow iron grading [25,38]. Unfortunately, the sensitivity of serum iron indices is rather diverse for detecting iron deficiency [25,38]; so, we categorized our cohort into low/normal iron groups or low/normal TSAT groups by utilizing the tertiles of serum iron or TSAT levels rather than using the traditional cutoff values of serum iron (70 μg/dL in men and 60 μg/dL in women) and TSAT (20%). Consequently, our study revealed that male patients within the first tertile of iron (<67 μg/dL) or within the first tertile of TSAT (<24.6%) had a higher risk of poor renal outcomes.

The relationship between serum iron and poor renal outcomes, especially for the male CKD patients, was postulated to be due to the more detrimental influence of muscle wasting, malnutrition, and inflammation in men, causing increased hepcidin production and decreased iron released to the blood, inducing decreased energy metabolism for high-energy-demand organs such as kidneys. Indeed, for the multivariate linear regression we performed (Appendix A), a negative correlation between MIS and serum iron in the men and women existed, but to varying extents. It showed that MIS had a significantly negative correlation with serum iron only among the male patients. We further tested the association between the interaction of sex with MIS and serum iron using a general linear model, and it revealed a significant association (βeta coefficient −0.034; 95% CI −0.066 to −0.001; *p* = 0.041). This sex difference could be explained as being the result of different body compositions and by the divergent impacts of inflammation among the sexes. With regard to body composition, physiologically men have a significantly higher skeletal muscle mass distribution than women [40,41] in both the general population [42] and CKD groups [41,43,44]. This different body constitution might be attributed to sex hormones. As one of the anabolic hormones, testosterone induces muscle hypertrophy by promoting the differentiation of myoblasts, enhancing amino acid utilization by skeletal muscle, and inhibiting apoptosis of muscle cells [45,46,47,48]. In contrast, female sex hormones are more related to the change in fat mass [49]. As muscle wasting is closely correlated to malnutrition–inflammation [50,51], male CKD patients tend to suffer from malnutrition–inflammation because they have higher muscle mass levels than females. Previous literature has shown that muscle mass is a more dominant prognostic factor for men, while fat mass might be more associated with the prognosis of women [43,52]. In addition to the different body conformations, the diverse implications of inflammation among the sexes might also play a role. Stenvinkel et al. report that elevated CRP is an important predictor for poor outcomes in male patients but not in females [52,53]. The immune modulation effect of estrogen [54] and a sex difference in the percentage of interleukin-1 positive mononuclear cells [55] might be the possible mechanisms. Altogether, we assumed that muscle wasting, malnutrition, and inflammation in the male subjects might contribute to markedly decreased serum iron bioavailability by upregulation of the iron regulatory hormone hepcidin [56]. As iron is essential for energy metabolism [3], it might induce unfavorable renal outcomes considering the high energy demand of the kidneys [16,17].

Our finding of the relationship between iron and renal outcomes in non-anemic CKD patients might highlight the momentousness of iron in bioenergetics beyond heme synthesis. Conversely, a lack of significance was found in the correlation between serum iron and renal outcomes among anemic CKD subjects, as revealed in Table 5 and Table 6. The mechanisms of anemia in CKD are multifactorial and include inhibited erythropoietin production by the kidney, shortened lifespan of red blood cells, iron deficiency, and chronic inflammation [37]. These multiple etiologies of anemia, except for iron deficiency, might have driven the irrelevance of the serum iron and renal outcomes in our anemic cohorts.

Furthermore, we found that both serum iron and TSAT were relevant to renal outcomes. Traditionally, TSAT and ferritin are the most frequently used biomarkers to assess iron stores and provide guidance for iron treatment to clinicians [33,35,36]. Nonetheless, serum ferritin, as an acute-phase protein, may become elevated in the settings of inflammation, malignancy, and liver disease [57,58]. In contrast, TSAT has better sensitivity than ferritin in terms of iron deficiency [59] while also having more directional correlations with mortality in CKD populations compared to ferritin [33,34,35]. Our study exhibited the fact that TSAT was closely associated with renal outcomes in CKD patients. However, TSAT increased when transferrin or TIBC decreased under malnutrition and inflammation [60,61,62]. In our previous study, we demonstrated that CKD stage 1–4 patients with low serum iron but normal TSAT were still associated with anemia [63]. Hence, we utilized serum iron to demonstrate the independent role of iron status in predicting renal outcomes. Additionally, our study further revealed that serum iron was also correlated with renal outcomes.

Several limitations of this study need to be mentioned. First, the observational inherence of our study could not allow us to establish a causal relationship between serum iron and the outcomes. Further well-designed interventional studies are required to confirm the association between iron levels and the renal outcomes in CKD populations. Second, the basic research to validate the mechanisms of renal function deterioration induced by iron deficiency was lacking. Third, the data on iron indices were obtained at a single time point, which is an impediment to discerning the effect of changes in iron markers on clinical outcomes over time. Fourth, the absence of the measurement of hepcidin and erythropoietin precludes the ability to interpret the state of iron availability and the capacity of hematopoiesis, respectively. Fifth, we did not examine the levels of sex hormones, which could strengthen the postulations of sex disparities in the relationship between serum iron and renal outcomes in this study. Sixth, our study included patients from an earlier cohort.

## 5. Conclusions

In conclusion, our study suggests that serum iron and TSAT are negatively correlated with poor renal outcomes (RRT or RRT + 50% decline in eGFR) in males and non-anemic patients with CKD stage 1–4. Both serum iron and TSAT were important predictors. This finding provides the potential implications for iron therapy and the augmentation of the screening of iron biomarkers in early-stage CKD as treatable risk factors of renal function deterioration. Further research is needed to validate our findings in terms of the impact of iron replacement treatment on renal outcomes.

## Figures and Tables

**Table 1 jpm-13-00521-t001:** Baseline characteristics of chronic kidney disease stage 1–4 patients divided by sex and serum iron.

Variables	Male			*p*-Value	Female			*p*-Value
	All	Low Iron	Normal Iron		All	Low Iron	Normal Iron	
		Iron Tertile 1 (<67 μg/dL)	Iron Tertiles 2 and 3 (≥67 μg/dL)			Iron Tertile 1 (<54 μg/dL)	Iron Tertiles 2 and 3 (≥54 μg/dL)	
Number of patients	1601	536	1065		899	299	600	
**Demographics, medical histories, and clinical parameters**								
Age (year)	62.8 (14.4)	63.5 (14.2)	62.4 (14.5)	0.20	61.7 (14.6)	61.5 (15.2)	61.8 (14.4)	0.78
DM (%)	764 (47.7%)	317 (59.1%)	447 (42.0%)	<0.001	466 (51.8%)	181 (60.5%)	285 (47.5%)	<0.001
CVD (%)	365 (22.8%)	150 (28.0%)	215 (20.2%)	<0.001	188 (20.9%)	68 (22.7%)	120 (20.0%)	0.34
HTN (%)	957 (59.8%)	334 (62.3%)	623 (58.5%)	0.142	556 (61.8%)	186 (62.2%)	370 (61.7%)	0.875
Hyperuricemia (%)	366 (22.9%)	112 (20.9%)	254 (23.8%)	0.184	82 (9.1%)	18 (6.0%)	64 (10.7%)	0.023
MtS (%)	1027 (64.1%)	357 (66.6%)	670 (62.9%)	0.146	646 (71.9%)	224 (74.9%)	422 (70.3%)	0.150
Malnutrition–inflammation (%)	370 (23.1%)	204 (38.1%)	166 (15.6%)	<0.001	231 (25.7%)	116 (38.8%)	115 (19.2%)	<0.001
MAP	99.5 (13.4)	99.4 (13.5)	99.6 (13.4)	0.814	99.1 (13.5)	101.3 (14.4)	97.9 (12.9)	<0.001
BMI	25.3 (3.8)	25.0 (4.2)	25.4 (3.6)	0.078	24.8 (4.4)	24.6 (4.5)	24.9 (4.3)	0.355
MIS	3.7 (2.9)	4.9 (3.2)	3.1 (2.5)	<0.001	4.0 (3.0)	4.8 (3.2)	3.6 (2.7)	<0.001
**Renal function status**								
eGFR (ml/min/1.73 m^2^)	38.2 (26.7–49.7)	34.9 (23.7–46.3)	40.1 (28.3–51.1)	<0.001	30.1 (21.7–43.7)	27.7 (20.4–42.8)	30.8 (22.4–44.2)	0.13
UPCR (mg/g)	575 (204–1521)	809 (250–2192)	488 (186–1267)	<0.001	1000 (346–2463)	1318 (395–3573)	815 (334–2184)	<0.001
CKD 1 + 2	213 (13.3%)	68 (12.7%)	145 (13.6%)	0.023	143 (15.9%)	61 (20.4%)	82 (13.7%)	0.035
CKD3	874 (54.6%)	273 (50.9%)	601 (56.4%)		309 (34.4%)	78 (26.1%)	231 (38.5%)	
CKD4	514 (32.1%)	195 (36.4%)	319 (30.0%)		447 (49.7%)	160 (53.5%)	287 (47.8%)	
**Laboratory data**								
Hb (g/dL)	12.8 (2.2)	12.0 (2.3)	13.2 (2.1)	<0.001	11.2 (1.8)	10.6 (1.8)	11.4 (1.7)	<0.001
Albumin (g/dL)	3.9 (0.5)	3.7 (0.6)	4.0 (0.5)	<0.001	3.9 (0.5)	3.7 (0.6)	4.0 (0.5)	<0.001
WBC (×1000 cells/μL)	7.2 (2.3)	7.8 (2.5)	7.0 (2.1)	<0.001	7.1 (2.3)	7.4 (2.5)	6.9 (2.2)	0.004
CRP (mg/L)	1.1 (0.3–4.7)	1.9 (0.5–8.3)	0.8 (0.3–3.4)	<0.001	0.9 (0.3–4.1)	1.1 (0.4–6.2)	0.8 (0.3–3.0)	0.004
Bicarbonate (mEq/L)	24.0 (3.6)	23.6 (3.9)	24.3 (3.5)	<0.001	23.1 (3.8)	23.0 (4.1)	23.1 (3.7)	0.723
Phosphorus (mg/dL)	3.7 (0.8)	3.8 (0.9)	3.7 (0.7)	0.002	4.2 (0.8)	4.2 (1.0)	4.2 (0.8)	0.962
Calcium (mg/dL)	9.2 (0.6)	9.2 (0.7)	9.3 (0.6)	0.056	9.3 (0.7)	9.2 (0.7)	9.3 (0.6)	0.143
Uric acid (mg/dL)	7.8 (1.9)	7.7 (1.9)	7.8 (1.9)	0.639	7.3 (2.1)	7.4 (2.3)	7.3 (1.9)	0.431
HbA1c (%)	6.6 (1.7)	7.0 (1.9)	6.4 (1.5)	<0.001	6.7 (1.8)	7.0 (2.1)	6.5 (1.6)	<0.001
**Outcomes**								
RRT	313 (19.6%)	137 (25.6%)	176 (16.5%)	<0.001	234 (26.0%)	84 (28.1%)	150 (25.0%)	<0.001
RRT and 50% decline in eGFR	432 (27.0%)	171 (31.9%)	261 (24.5%)	<0.001	349 (38.8%)	126 (42.1%)	223 (37.2%)	0.41
Mortality before RRT	274 (17.1%)	124 (23.1%)	150 (14.1%)	<0.001	136 (15.1%)	49 (16.4%)	87 (14.5%)	<0.001

Abbreviations: DM, diabetes mellitus; CVD, cardiovascular disease; HTN, hypertension; MtS, metabolic syndrome; MAP, mean arterial blood pressure; BMI, body mass index; MIS, malnutrition–inflammation score; eGFR, estimated glomerular filtration rate; UPCR, urine protein to creatinine ratio; CKD, chronic kidney disease; Hb, hemoglobin; WBC, white blood cell count; CRP, C-reactive protein; HbA1c, glycated hemoglobin; RRT, renal replacement therapy.

**Table 2 jpm-13-00521-t002:** Iron status divided by sex and serum iron in chronic kidney disease stage 1–4 patients.

Variables	Male			*p*-Value	Female			*p*-Value
	All	Low Iron	Normal Iron		All	Low Iron	Normal Iron	
		Iron Tertile 1 (<67 μg/dL)	Iron Tertiles 2 and 3 (≥67 μg/dL)			Iron Tertile 1 (<54 μg/dL)	Iron Tertiles 2 and 3 (≥54 μg/dL)	
Number of patients	1601	536	1065		899	299	600	
Serum iron (μg/dL)	83.2 (37.6)	46.4 (17.0)	101.8 (30.8)	<0.001	68.0 (32.3)	35.4 (14.7)	84.3 (25.7)	<0.001
TSAT (%)	31.6 (15.8)	20.3 (11.6)	37.3 (14.6)	<0.001	27.0 (15.9)	16.7 (11.9)	32.1 (15.2)	<0.001
TIBC (μg/dL)	276.2 (78.1)	285.0 (70.0)	261.4 (88.2)	<0.001	277.4 (87.3)	286.4 (79.5)	265.3 (95.7)	<0.001
Ferritin (ng/mL)	208.9 (119.4–355.5)	190.7 (98.8–356.6)	214.9 (129.6–354.8)	0.009	131.8 (63.3–236.4)	109.7 (40.0–235.1)	138.8 (80.8–236.6)	<0.001
Albumin < 3.5 g/dL (%)	273 (17.1%)	149 (27.8%)	124 (11.6%)	<0.001	177 (19.7%)	86 (28.8%)	91 (15.2%)	<0.001
CRP > 3 mg/L (%)	502 (31.4%)	220 (41.0%)	282 (26.5%)	<0.001	254 (28.3%)	106 (35.5%)	148 (24.7%)	<0.001
TIBC < 200 μg/dL (%)	241 (15.1%)	111 (20.7%)	130 (12.2%)	<0.001	156 (17.4%)	72 (24.1%)	84 (14.0%)	0.004
Ferritin > 200 ng/mL (%)	831 (51.9%)	255 (47.6%)	576 (54.1%)	0.014	284 (31.6%)	91 (30.4%)	193 (32.2%)	0.599
Hb < 11 g/dL (%)	365 (22.8%)	186 (34.7%)	179 (16.8%)	<0.001	441 (49.1%)	180 (60.2%)	261 (43.5%)	<0.001
MIS > 4 points (%)	370 (23.1%)	204 (38.1%)	166 (15.6%)	<0.001	231 (25.7%)	116 (38.8%)	115 (19.2%)	<0.001

Abbreviations: TSAT, transferrin saturation; TIBC, total iron-binding capacity; CRP, C-reactive protein; Hb, hemoglobin; MIS, malnutrition–inflammation score.

**Table 3 jpm-13-00521-t003:** Association of serum iron with renal outcomes divided by sex.

	Male			Female		
	Per 1 SD Increase in Iron	Low Iron	Normal Iron	Per 1 SD Increase in Iron	Low Iron	Normal Iron
		Iron Tertile 1 (<67 μg/dL)	Iron Tertiles 2 and 3 (≥67 μg/dL)		Iron Tertile 1 (<54 μg/dL)	Iron Tertiles 2 and 3 (≥54 μg/dL)
**HR (95% CI) for RRT**						
Unadjusted	0.653 (0.575–0.740) *	1 (reference)	0.477 (0.381–0.597) *	0.847 (0.743–0.966) *	1 (reference)	0.751 (0.575–0.981) *
Fully adjusted	0.838 (0.717–0.980) *	1 (reference)	0.640 (0.500–0.820) *	1.191 (0.985–1.440)	1 (reference)	1.061 (0.779–1.447)
**HR (95% CI) for RRT + 50% decline of eGFR**						
Unadjusted	0.714 (0.642–0.792) *	1 (reference)	0.562 (0.463–0.682) *	0.842 (0.757–0.936) *	1 (reference)	0.747 (0.600–0.929) *
Fully adjusted	0.882 (0.778–0.998) *	1 (reference)	0.718 (0.579–0.889) *	1.030 (0.889–1.193)	1 (reference)	0.940 (0.732–1.205)

Abbreviations: SD, standard deviation; HR, hazard ratio; CI, confidence interval; RRT, renal replacement therapy; eGFR, estimated glomerular filtration rate. * Means statistically significant compared with reference group. Adjusted covariates: age, eGFR, log-transformed urine protein to creatinine ratio, diabetes mellitus, cardiovascular disease, severe liver disease, cancer, mean blood pressure, body mass index, albumin, log-transformed C-reactive protein, hemoglobin, log-transformed ferritin, phosphorus, total cholesterol, and glycated hemoglobin.

**Table 4 jpm-13-00521-t004:** Association of TSAT with renal outcomes divided by sex.

	Male			Female		
	Per 1 SD Increase in TSAT	Low TSAT	Normal TSAT	Per 1 SD Increase in TSAT	Low TSAT	Normal TSAT
		TSAT Tertile 1 (<24.6%)	TSAT Tertiles 2 and 3 (≥24.6%)		TSAT Tertile 1 (<19.7%)	TSAT Tertiles 2 and 3 (≥19.7%)
**HR (95% CI) for RRT**						
Unadjusted	0.797 (0.703–0.903) *	1 (reference)	0.644 (0.513–0.808) *	0.958 (0.840–1.091)	1 (reference)	1.124 (0.850–1.487)
Fully adjusted	0.847 (0.734–0.978) *	1 (reference)	0.662 (0.515–0.852) *	1.122 (0.947–1.330)	1 (reference)	1.085 (0.785–1.499)
**HR (95% CI) for RRT + 50% decline in eGFR**						
Unadjusted	0.835 (0.752–0.926) *	1 (reference)	0.725 (0.596–0.882) *	0.954 (0.859–1.059)	1 (reference)	1.091 (0.870–1.369)
Fully adjusted	0.870 (0.774–0.978) *	1 (reference)	0.770 (0.620–0.957) *	1.047 (0.917–1.195)	1 (reference)	1.099 (0.848–1.424)

Abbreviations: TSAT, transferrin saturation; SD, standard deviation; HR, hazard ratio; CI, confidence interval; RRT, renal replacement therapy; eGFR, estimated glomerular filtration rate. * Means statistically significant compared with reference group. Adjusted covariates: age, eGFR, log-transformed urine protein to creatinine ratio, diabetes mellitus, cardiovascular disease, severe liver disease, cancer, mean blood pressure, body mass index, albumin, log-transformed C-reactive protein, hemoglobin, log-transformed ferritin, phosphorus, total cholesterol, and glycated hemoglobin.

**Table 5 jpm-13-00521-t005:** Association of serum iron with renal outcomes divided by hemoglobin.

	Hb < 11 g/dL			Hb ≥ 11 g/dL		
	Per 1 SD Increase in Iron	Low Iron	Normal Iron	Per 1 SD Increase in Iron	Low Iron	Normal Iron
		Iron Tertile 1 (Men < 67 μg/dL) (Women < 54 μg/dL)	Iron Tertiles 2 and 3 (Men ≥ 67 μg/dL) (Women ≥ 54 μg/dL)		Iron Tertile 1 (Men < 67 μg/dL) (Women < 54 μg/dL)	Iron Tertiles 2 and 3 (Men ≥ 67 μg/dL) (Women ≥ 54 μg/dL)
**HR (95% CI) for RRT**						
Unadjusted	0.955 (0.835–1.093)	1 (reference)	0.817 (0.650–1.027)	0.708 (0.620–0.808) *	1 (reference)	0.557 (0.430–0.722) *
Fully adjusted	1.130 (0.972–1.313)	1 (reference)	0.943 (0.730–1.218)	0.821 (0.687–0.980) *	1 (reference)	0.699 (0.525–0.929) *
**HR (95% CI) for RRT + 50% decline in eGFR**						
Unadjusted	0.952 (0.845–1.073)	1 (reference)	0.848 (0.693–1.039)	0.775 (0.700–0.857) *	1 (reference)	0.654 (0.530–0.806) *
Fully adjusted	1.076 (0.940–1.233)	1 (reference)	0.952 (0.761–1.191)	0.854 (0.749–0.975) *	1 (reference)	0.715 (0.568–0.898) *

Abbreviations: Hb, hemoglobin; SD, standard deviation; HR, hazard ratio; CI, confidence interval; RRT, renal replacement therapy; eGFR, estimated glomerular filtration rate. * Means statistically significant compared with reference group. Adjusted covariates: age, eGFR, log-transformed urine protein to creatinine ratio, diabetes mellitus, cardiovascular disease, severe liver disease, cancer, mean blood pressure, body mass index, albumin, log-transformed C-reactive protein, log-transformed ferritin, phosphorus, total cholesterol, and glycated hemoglobin.

**Table 6 jpm-13-00521-t006:** Association of TSAT with renal outcomes divided by hemoglobin.

	Hb < 11 g/dL			Hb ≥ 11 g/dL		
	Per 1 SD Increase in TSAT	Low TSAT	Normal TSAT	Per 1 SD Increase in TSAT	Low TSAT	Normal TSAT
		TSAT Tertile 1 (men < 24.6%) (women < 19.7%)	TSAT Tertiles 2 and 3 (Men ≥ 24.6%) (Women ≥ 19.7%)		TSAT Tertile 1 (Men < 24.6%) (Women < 19.7%)	TSAT Tertiles 2 and 3 (Men ≥ 24.6%) (women ≥ 19.7%)
**HR (95% CI) for RRT**						
Unadjusted	1.032 (0.913–1.166)	1 (reference)	1.081 (0.852–1.371)	0.785 (0.682–0.902) *	1 (reference)	0.697 (0.538–0.903) *
Fully adjusted	1.053 (0.915–1.211)	1 (reference)	0.962 (0.730–1.268)	0.841 (0.708–0.999) *	1 (reference)	0.752 (0.560–1.011)
**HR (95% CI) for RRT + 50% decline in eGFR**						
Unadjusted	1.038 (0.933–1.155)	1 (reference)	1.101 (0.892–1.358)	0.836 (0.753–0.929) *	1 (reference)	0.803 (0.652–0.989) *
Fully adjusted	1.070 (0.946–1.211)	1 (reference)	1.052 (0.826–1.341)	0.858 (0.756–0.973) *	1 (reference)	0.796 (0.630–1.006)

Abbreviations: Hb, hemoglobin; TSAT, transferrin saturation; SD, standard deviation; HR, hazard ratio; CI, confidence interval; RRT, renal replacement therapy; eGFR, estimated glomerular filtration rate. * Means statistically significant compared with reference group. Adjusted covariates: age, eGFR, log-transformed urine protein to creatinine ratio, diabetes mellitus, cardiovascular disease, severe liver disease, cancer, mean blood pressure, body mass index, albumin, log-transformed C-reactive protein, log-transformed ferritin, phosphorus, total cholesterol, and glycated hemoglobin.

**Table 7 jpm-13-00521-t007:** Association of “combined TSAT and serum iron” with renal outcomes among male chronic kidney disease stage 1–4 patients.

	Male CKD Stage 1–4 Patients			
	Low Iron/ Low TSAT	Low Iron/ Normal TSAT	Normal Iron/ Low TSAT	Normal Iron/ Normal TSAT
**HR (95% CI) for RRT**				
Unadjusted	1 (reference)	1.288 (0.898–1.848)	0.659 (0.437–0.995) *	0.491 (0.378–0.636) *
Fully adjusted	1 (reference)	1.141 (0.777–1.676)	1.161 (0.754–1.788)	0.603 (0.452–0.804) *
**HR (95% CI) for RRT + 50% decline in eGFR**				
Unadjusted	1 (reference)	1.380 (1.000–1.905)	0.764 (0.538–1.084)	0.593 (0.473–0.742) *
Fully adjusted	1 (reference)	1.322 (0.942–1.855)	1.126 (0.782–1.620)	0.713 (0.556–0.914) *

Abbreviations: CKD, chronic kidney disease; TSAT, transferrin saturation; SD, standard deviation; HR, hazard ratio; CI, confidence interval; RRT, renal replacement therapy; eGFR, estimated glomerular filtration rate. * Means statistically significant compared with reference group. Adjusted covariates: age, eGFR, log-transformed urine protein to creatinine ratio, diabetes mellitus, cardiovascular disease, severe liver disease, cancer, mean blood pressure, body mass index, albumin, log-transformed C-reactive protein, hemoglobin, log-transformed ferritin, phosphorus, total cholesterol, and glycated hemoglobin.

## Data Availability

Study data are available from the corresponding author (C.-C.H.) upon request.

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
