# Peer review of "The Association between Iron Deficiency and Renal Outcomes Is Modified by Sex and Anemia in Patients with Chronic Kidney Disease Stage 1–4"

_jpm, 2023, doi:10.3390/jpm13030521_

Round 1
Reviewer 1 Report
lines 77-78: ..... enrolled from the Integrated CKD Care Program Kaohsiung for Delaying Dialysis 77 from 11 November, 2002 to 31 May, 2009 .......
- Comment: why the study included patients from so many years ago? This is very unusual and authors should explain this.
lines 142-143: ... Ferritin was measured in serum using two-site im- 142 muno-enzymatic assay by Beckman Coulter UniCel-DxI 800 (Beckman Coulter. ...
- Comment: measurement of ferritin definitely makes sense but why ferritin is not presented several lines earlier among studied laboratory data?
lines 190-195 - Comment: data presented in the text are presented in Table 1 as well. Duplication of data presentation should be avoided.
lines 217-221 - Comment: data presented in the text are presented in Table 2 as well. Duplication of data presentation should be avoided.
Author Response
Dear Editors and Reviewers:
We highly appreciate the thoughtful comments by the Editors and the Reviewers, which enabled us to clarify and improve our manuscript. We hope that the given answers and the changes we have made are sufficient for publication of the manuscript in Journal of Personalized Medicine.
Sincerely yours,
Chi-Chih Hung, on behalf of other co-authors
Ponit 1: lines 77-78: ..... enrolled from the Integrated CKD Care Program Kaohsiung for Delaying
Dialysis 77 from 11 November, 2002 to 31 May, 2009 .......
- Comment: why the study included patients from so many years ago? This is very unusual and authors should explain this.
Response 1:
Thanks for your comments. We have added this shortcoming as “the sixth point” in the part of limitation. In fact, we have applied this cohort to explore several risk factors of poor renal outcomes, including of metabolic syndrome, HbA1c, hyperuricemia, body mass index, and so on in the past two years[1-4]. We also published a paper in 2021, which investigated the association of serum iron and anemia in CKD stage 1-4 populations[5]. Moreover, owing to the historical background, erythropoiesis-stimulating agents (ESAs) were not applied in our cohort because there’s no reimbursement by national health insurance at that time. (The criteria of ESAs treatment by national health insurance of Taiwan before December 1, 2015: CKD patients with creatinine >6 mg/dL and hematocrit ≦28%). ESAs may be a significant confounding factor of the relationship between serum iron and anemia. Without both confounders of ESAs and anemia, the association of serum iron with renal outcomes might be strengthened.
Reference:
1. Lin, H.Y.; Chang, L.Y.; Niu, S.W.; Kuo, I.C.; Yen, C.H.; Shen, F.C.; Chen, P.L.; Chang, J.M.; Hung, C.C. High risk of renal outcome of metabolic syndrome independent of diabetes in patients with ckd stage 1-4: The ickd database. Diabetes Metab Res Rev 2023, e3618.
2. Hung, C.C.; Zhen, Y.Y.; Niu, S.W.; Lin, K.D.; Lin, H.Y.; Lee, J.J.; Chang, J.M.; Kuo, I.C. Predictive value of hba1c and metabolic syndrome for renal outcome in non-diabetic ckd stage 1-4 patients. Biomedicines 2022, 10.
3. Niu, S.W.; Lin, H.Y.; Kuo, I.C.; Zhen, Y.Y.; Chang, E.E.; Shen, F.C.; Chiu, Y.W.; Chang,
J. M.; Hung, C.C.; Hwang, S.J. Hyperuricemia, a non-independent component of metabolic syndrome, only predicts renal outcome in chronic kidney disease patients without metabolic syndrome or diabetes. Biomedicines 2022, 10.
4. Hung, C.C.; Yu, P.H.; Niu, S.W.; Kuo, I.C.; Lee, J.J.; Shen, F.C.; Chang, J.M.; Hwang, S.J. Association between body mass index and renal outcomes modified by chronic kidney disease and anemia: The obesity paradox for renal outcomes. J Clin Med 2022, 11.
5. Yu, P.H.; Lin, M.Y.; Chiu, Y.W.; Lee, J.J.; Hwang, S.J.; Hung, C.C.; Chen, H.C. Low serum iron is associated with anemia in ckd stage 1-4 patients with normal transferrin saturations. Sci Rep 2021, 11, 8343.
Point 2: lines 142-143: ... Ferritin was measured in serum using two-site im- 142 muno-enzymatic assay by Beckman Coulter UniCel-DxI 800 (Beckman Coulter. ...
- Comment: measurement of ferritin definitely makes sense but why ferritin is not presented several lines earlier among studied laboratory data?
Response 2:
Thanks for your comments. We have presented “the measurement of ferritin” several lines earlier among studied laboratory data in the part of “2.2. Demographic, medical, lab data, and clinical parameter collection”. Actually, the past literatures have argued that the accuracy of ferritin because its concentration may increase in the situation of inflammation and malnutrition[6,7]. And there’s a certain proportion of malnutrition-inflammation disorders among CKD patients. Professor Jay B. Wish indicated that TSAT has better sensitivity than ferritin in terms of iron deficiency[8]. Furthermore, Professor Besarab and Professor Drueke mentioned that serum iron may provide more information to guide iron therapy than TSAT because TSAT is misplaced in the setting of inflammation since its denominator- transferrin is decreased via proinflammatory cytokine action in the liver[9]. In our previous study, we also revealed that CKD stage 1-4 patients with low serum iron but normal TSAT were still associated with anemia[5]. Therefore, we utilized both serum iron and TSAT as the main iron biomarkers to predict renal outcomes in the present study.
Reference:
5. Yu, P.H.; Lin, M.Y.; Chiu, Y.W.; Lee, J.J.; Hwang, S.J.; Hung, C.C.; Chen, H.C. Low serum iron is associated with anemia in ckd stage 1-4 patients with normal transferrin saturations. Sci Rep 2021, 11, 8343.
6. Lopez, A.; Cacoub, P.; Macdougall, I.C.; Peyrin-Biroulet, L. Iron deficiency anaemia. The Lancet 2016, 387, 907-916.
7. Kalantar-Zadeh, K.; Kalantar-Zadeh, K.; Lee, G.H. The fascinating but deceptive ferritin: To measure it or not to measure it in chronic kidney disease? Clin J Am Soc Nephrol 2006, 1 Suppl 1, S9-18.
8. Wish, J.B. Assessing iron status: Beyond serum ferritin and transferrin saturation. Clin J Am Soc Nephrol 2006, 1 Suppl 1, S4-8.
9. Besarab, A.; Drueke, T.B. The problem with transferrin saturation as an indicator of iron 'sufficiency' in chronic kidney disease. Nephrol Dial Transplant 2021, 36, 1377-1383.
Point 3: lines 190-195 - Comment: data presented in the text are presented in Table 1 as well. Duplication of data presentation should be avoided.
Response 3: Thanks for your comments. We have made correction in the revised manuscript.
Point 4: lines 217-221 - Comment: data presented in the text are presented in Table 2 as well. Duplication of data presentation should be avoided.
Response 3: Thanks for your suggestions. We have made revision according to the comments.

Reviewer 2 Report
In the MS, Dr. Yu and co-authors explore the relationship between iron status and renal outcomes as well as investigate the impact of sex and anemia in this association among patients with CKD stage 1-4. They follow up with 2500 CKD patients of 8.2 years and found that serum iron and TSAT were negatively correlated with poor renal outcomes in males and non- anemic patients with CKD stage 1-4. However, female patients did not show any significant relationship between serum iron and TSAT and renal outcomes. This finding provides potential implications for iron therapy and augmentation of screening of iron biomarkers in early-stage CKD as treatable risk factors of renal function deterioration. It is interesting research and has clinical value. I like authors to address these questions.
1. In MS, the authors stratified these 2,500 CKD subjects into four groups - “men with normal iron”, “men with low iron”, “women with normal iron”, and “women with low iron” in accordance with the tertiles of serum iron levels. The authors need to add a column to show the population size of each stage in low and normal iron cohorts. Otherwise, it might cause some misunderstanding of the role of low iron in accelerating CKD progression in the clinic.
2. The MS showed that iron deficiency is associated with poor renal outcomes, summarizing and discussing the mechanisms of renal function deterioration as induced by iron deficiency would be better.
Author Response
Dear Editors and Reviewers:
We highly appreciate the thoughtful comments by the Editors and the Reviewers, which enabled us to clarify and improve our manuscript. We hope that the given answers and the changes we have made are sufficient for publication of the manuscript in Journal of Personalized Medicine.
Sincerely yours,
Chi-Chih Hung, on behalf of other co-authors
Point 1: In MS, the authors stratified these 2,500 CKD subjects into four groups - “men with normal iron”, “men with low iron”, “women with normal iron”, and “women with low iron” in accordance with the tertiles of serum iron levels. The authors need to add a column to show the population size of each stage in low and normal iron cohorts. Otherwise, it might cause some misunderstanding of the role of low iron in accelerating CKD progression in the clinic.
Response 1:
Thanks for your comments. We have added columns to disclose the population size of each CKD stage in low and normal iron cohort in Table 1. Besides, in our statistical analysis, we took eGFR as one of the adjusted covariates to mitigate the impact of baseline renal function on the relationship between iron status and renal outcomes.
Point 2: The MS showed that iron deficiency is associated with poor renal outcomes, summarizing and discussing the mechanisms of renal function deterioration as induced by iron deficiency would be better.
Response 2:
Our study presented that serum iron and TSAT were negatively correlated with poor renal outcomes in males and non-anemic patients with CKD stage 1-4. The relationship between serum iron and poor renal outcomes especially for male CKD patients was postulated due to more detrimental influence of muscle wasting, malnutrition and inflammation on men, which further increased hepcidin production and decreased irons released to the blood, inducing decreased energy metabolism for the high energy demand organ such as kidney. This hypothesis needs advanced studies to prove, though. And our finding of the relationship between iron and poor renal outcomes in non-anemic CKD patients just highlight the importance of iron in bioenergetics beyond heme synthesis because multiple etiologies of anemia except for iron deficiency might cause the irrelevance of serum iron and renal outcomes in anemic subjects. Thanks for your comment. We have revised the fourth and the fifth paragraphs of discussion to illustrate the possible mechanism more clearly.

Round 2
Reviewer 2 Report
Thanks for the authors' response. The manuscript significantly improved after revision; I have no additional comments.
Author Response
Dear Reviewer:
Thanks for your comments. We hope that the revised manuscript is sufficient for publication in Journal of Personalized Medicine.
Sincerely yours,
Chi-Chih Hung, on behalf of other co-authors